# Biological Regulation of HIF-1α and Its Role in Therapeutic Angiogenesis for Treatment of Ischemic Cardiovascular Disease

**DOI:** 10.3390/ijms262211236

**Published:** 2025-11-20

**Authors:** Ethan Carmichael, Anne-Isabelle S. Reme, Patrick J. Bosco, Yulexi Y. Ortiz, Daniela Alexandra Ramos, Katherine Gomez, Bao-Ngoc Nguyen, Arash Bornak, Zhao-Jun Liu, Omaida C. Velazquez

**Affiliations:** 1DeWitt Daughtry Family Department of Surgery, University of Miami Miller School of Medicine, Miami, FL 33136, USA; exc2834@med.miami.edu (E.C.); asr2250@miami.edu (A.-I.S.R.); pjb132@med.miami.edu (P.J.B.); yyo2@med.miami.edu (Y.Y.O.); dar420@miami.edu (D.A.R.); kgome107@fiu.edu (K.G.); abornak@med.miami.edu (A.B.); 2Medical Scientist Training Program, University of Miami Miller School of Medicine, Miami, FL 33136, USA; 3Department of Biochemistry & Molecular Biology, University of Miami Miller School of Medicine, Miami, FL 33136, USA; 4Department of Surgery, The George Washington University School of Medicine and Health Sciences, Washington DC 20037, USA; bnnguyen@mfa.gwu.edu; 5Department of Radiology, University of Miami Miller School of Medicine, Miami, FL 33136, USA

**Keywords:** HIF-1α, oxygen, hypoxia, ischemia, cardiovascular diseases

## Abstract

Hypoxia, characterized by insufficient oxygen saturation, triggers a wide array of vascular responses aimed at enhancing cell survival and proliferation. This process is primarily driven by the activation of oxygen-sensing hypoxia-inducible factors (HIFs). HIF-1α, a key mediator in this context, plays a crucial role in vascular restructuring in response to low oxygen tension and oxygen-independent signaling pathways, making it a promising therapeutic target for ischemic cardiovascular diseases such as peripheral artery disease and coronary artery disease. In this review, we explore both oxygen-dependent and oxygen-independent mechanisms of HIF-1α regulation, the role of the HIF protein family in vessel collateralization, and translational efforts to leverage HIF-1α‘s pivotal role in hypoxia signaling for the development of clinical treatments for ischemic cardiovascular disease.

## 1. Introduction

Oxygen homeostasis, the balance of oxygen supply and demand, is crucial to the normal functioning of molecular and cellular processes involved in cell metabolism, differentiation, proliferation, and survival, as well as organ function and human survival [1,2]. Humans have adapted cellular and biochemical responses to combat hypoxic insult key to many disease processes. Among these, hypoxia-inducible factors (HIFs) are the most extensively studied [3]. HIF-1 was identified in 1992 as a transcription factor that upregulates erythropoietin (EPO) production in response to hypoxia by binding to the EPO enhancer and increases its transcription [4,5,6]. EPO is a glycoprotein hormone produced by specialized interstitial peritubular fibroblast-like cells of the kidney which acts to promote erythropoiesis in the bone marrow in response to hypoxia and/or anemia, thereby increasing the number of circulating red blood cells and increasing oxygen delivery to tissues [7].

Further studies have since uncovered the hypoxic regulatory mechanisms of HIF-1α and the more than 100 genes that it regulates [8]. HIF-1α plays a crucial role in promoting the formation of new blood vessels (angiogenesis) through upregulation of growth factors such as VEGF [9,10,11], facilitating an energy-conserving metabolic switch from aerobic to anaerobic metabolism via the upregulation of key glycolytic enzymes, increase in glucose transporters in the cell membrane, and repression of mitochondrial TCA cycle enzymes—all of which effectively increase the intracellular oxygen tension [12,13]. Additionally, HIF-1α induction has been shown to increase EPO production to increase circulating RBC volume [7], dampen the inflammatory response via extracellular adenosine signaling [14], and promote cell proliferation and survival in hypoxic environments, including solid tumors [15,16].

These HIF-driven cellular mechanisms are central to tissue survival in response to ischemic events, such as the growth of new collateral blood vessels in occluded cerebral, peripheral, and coronary arteries to restore local circulation [17]. The purpose of this review paper is to explore the biological regulation of the HIF proteins, the role of HIFs and their downstream targets in promoting angiogenesis, and the clinical implications of therapeutic angiogenesis in ischemic coronary and peripheral artery disease.

## 2. Oxygen-Dependent Regulation of the HIFs

HIF-1α and HIF-2α are heterodimeric transcription factors that belong to the basic helix-loop-helix PER-ARNT-SIM family (bHLH-PAS). They consist of an oxygen-sensitive alpha subunit and a constitutively expressed beta subunit (HIF-1β) [18]. The beta subunit is also known as the aryl hydrocarbon receptor nuclear translocator (ARNT) and is encoded by ARNT1 and ARNT2 [19]. HIF-1β forms a heterodimer with both HIF-1α and HIF-2α [18]. There are three isoforms of the alpha subunit, named HIF-1α, HIF-2α, and HIF-3α, respectively [20]. While the alpha subunits of HIF-1 and HIF-2 exhibit stable transcription, they are tightly regulated at the protein level [21]. The alpha subunit contains an oxygen-dependent degradation (ODD) domain with two specific proline residues that undergo hydroxylation by several prolyl hydroxylase domain proteins (PHD1-4) under normal oxygen tension, or “normoxic” conditions [22]. Hydroxylation of the HIF alpha subunits occurs in the cytoplasm, leading to the binding of the alpha subunit to Von Hippel Lindau protein (VHL) [23]. This interaction forms a complex with the E3 ubiquitin ligase, resulting in polyubiquitylation and subsequent degradation via the ubiquitin-proteasome pathway [24]. The half-life of the HIF alpha subunits in the cytosol is approximately 5 min, leading to rapid protein degradation in normoxic conditions [25] (left side of Figure 1).

The hydroxylation of the HIF alpha subunit by PHD is contingent upon the presence of molecular oxygen, α-ketoglutarate, ascorbate, as well as iron as a catalyst [26]. Disruption of the iron catalyst, either through iron chelation with deferoxamine (DFO) or by competing with the PHD iron binding site using cobalt chloride (CoCl2), prevents PHD-mediated hydroxylation of HIF alpha subunits, thereby chemically stabilizing them in in vitro experiments [27,28]. Under hypoxic conditions, PHD proteins are unable to hydroxylate the HIF alpha subunits, allowing the HIF alpha subunit to translocate to the nucleus, where it forms a heterodimer with the HIF beta subunit [29]. The HIF heterodimer complex binds to specific core DNA sequences, typically located near the promoters of HIF target genes, known as hypoxic response elements (*HREs*) [30]. The bHLH sequence is essential for DNA-binding, while the three PAS regions, PAS-A, PAS-B, and PAS-associated C-terminal domain, are involved in heterodimerization [18]. HIF-1α and HIF-2α contain N-terminal and C-terminal transactivation domains (N-TAD and C-TAD, respectively) that are involved in the activation of HIF target genes [31]. These domains interact with additional transcriptional co-activators, most notably CBP and p300, which have lysine acetyl-transferase activity [32]. The CTAD region of the HIF alpha subunit polypeptide is subject to an additional level of oxygen-dependent regulation via the Factor Inhibiting HIF (FIH). FIH hydroxylates an asparagine residue within the CTAD domains at even lower oxygen tensions than PHD proteins, due to its lower Km for oxygen, thus exerting negative regulation on the HIF alpha subunits even under hypoxic conditions [33]. 

HIF-1α and HIF-2α exhibit strong sequence conservation between their bHLH and PAS regions, indicating their ability to bind identical DNA regions [18]. However, their NTAD regions confer target gene selectivity to the two proteins, likely secondary to distinct interactions with various transcriptional co-activators [34]. Interestingly, the CTAD region shows the least sequence conservation between the two proteins, yet they act to transactivate genes common to both HIF-1α and HIF-2α [35]. Several splice variants of HIF-3α exist, lacking a functional CTAD region and possibly an NTAD region [20]. The most extensively studied variant, HIF-3AF, lacks both transactivation domains, and functions to negatively regulate HIF-1α in an oxygen-independent manner [36]. HIF-3α and its therapeutic potential for neovascularization are less explored in the literature. Therefore, the remainder of this review will focus on HIF-1α and HIF-2α.

HIF-1α and HIF-2α demonstrate temporal differences in their gene expression, with HIF-1α responding to acute hypoxia within minutes and rapidly inducing the expression of its downstream target genes [37,38]. At around 8 h, HIF-1α levels peak and begin to decrease and HIF-2α levels begin to rise. By 24–48 h, HIF-2α levels become the more active responder to chronic hypoxia [34,39]. The fall in HIF-1α protein levels can be partly attributed to hypoxia-associated factor (HAF) mediated ubiquitination which targets HIF-1α for VHL-mediated protein degradation in proliferating cells irrespective of oxygen tension [40]. This oxygen-independent regulation does not apply to HIF-2α [41].

## 3. Oxygen-Independent Regulation of the HIFs

Although the HIF alpha subunits are traditionally regulated at the post-translational level through oxygen-dependent hydroxylation, they are also influenced by oxygen-independent interactions with other cell signaling pathways [42]. One area of signaling crosstalk comes from the NF-κB pathway [43]. It is unsurprising that crosstalk exists between hypoxia and inflammatory cell signaling. Indeed, several studies have identified an NF-κB binding site within the promoter of HIF-1α [44]. One such study demonstrated that HIF-1α mRNA and protein levels increased in response to exogenous reactive oxygen species (ROS) administration, specifically H_2_O_2_, to cultured pulmonary artery smooth muscle cells (PASMCs) under normoxic conditions [44,45]. This finding suggests that NF-κB directly upregulates HIF-1α transcription in a manner that does not depend on oxygen levels [46].

Interestingly, TNF-α, a potent cell surface activator of NF-κB, has been shown to promote HIF-1α protein activity. However, the mechanism remains debated due to conflicting findings regarding increased HIF-1α DNA binding, elevated HIF-1α mRNA levels, and post-translational protein stabilization that may vary based on cell types and experimental conditions [47,48,49]. For example, TNF-α has been shown to upregulate HIF-1α mRNA and protein levels via NF-κB in human pterygium fibroblasts in normoxic conditions [50] while it interferes with the transcription of HIF target genes in cultured smooth muscle cells during hypoxia [51]. IL-1β can also upregulate HIF-1α in an NF-κB-dependent manner [52]. Additionally, NF-κB can enhance HIF-1α expression in hypoxic environments, particularly in the central regions of the solid tumor microenvironment [53]. Hypoxia has been shown to upregulate HIF-1α mRNA via NF-κB through a PI3K/AKT pathway dependent mechanism in PASMCs [54].

The PI3K/AKT/mTOR and PI3K/AKT/FRAP pathways can also induce HIF-1α expression independently of NF-κB, in both normoxia and hypoxia [55]. Various cell surface ligands and receptors, such as EGFR, PDGF, TNF-α, IL-1β, and insulin, can activate the PI3K/AKT pathway [9,56]. Interestingly, recent studies have identified growth hormone-releasing hormone (GHRH) as an upstream, oxygen-independent activator of HIF-1α in iPSC-derived cardiomyocytes via GHRH/GHRH-R/cAMP signaling, acting as a mediator of cardiomyocyte proliferation and oxidative phosphorylation [57,58].

Additionally, various post-translational modifications can occur in HIF proteins, such as the phosphorylation and acetylation of the HIF-1α protein [59]. These modifications can either positively or negatively regulate HIF-1α, depending on the location of the modified amino acid within the protein [60]. For instance, the phosphorylation of serine residues by *ERK1/2* in the MAPK pathway enhances HIF-1α transcriptional activity and promotes cell survival following hypoxic injury in cardiomyocytes [61]. Conversely, phosphorylation events in the PAS or ODD regions inhibit HIF-1α protein activity [62].

Moreover, the post-transcriptional modification of HIF-1α mRNA by microRNAs (miRNAs) introduces an additional layer of regulation [63]. Active HIF-1α directly upregulates several small ~22 bp miRNAs, which in turn regulate the stability of HIF-1α mRNA or protein, either positively or negatively [64]. Among these, miR-210 is the most extensively studied miRNA involved in regulating HIF-1α activity [65,66,67]. Under hypoxic conditions, HIF-1α directly upregulates miR-210, which then binds to its target protein glycerol-3-phosphate dehydrogenase 1-like (GPD1L) [68]. GPD1L normally enhances the activity of PHD enzymes, promoting HIF-1α protein hydroxylation and degradation [68]. The increase in miR-210 by active HIF-1α during hypoxia creates a positive feedback loop leading to the downregulation of GPD1L, reduced PHD enzyme activity, and stabilized HIF-1α protein [65,69]. Other miRNAs, such as miR-155, can bind to the 3′ UTR region of HIF-1α mRNA transcripts, interfering with translation [70] (right side of Figure 1).

## 4. HIF Proteins and Neovascularization

The term “neovascularization” encompasses the various processes that lead to the formation of new blood vessels, including vasculogenesis, arteriogenesis, and angiogenesis [71]. Vasculogenesis occurs during embryonic development and involves de novo formation of blood vessels from vascular progenitor cells [72]. Although HIF proteins play a pivotal role in vasculogenesis, this process is confined to embryonic development and is beyond the scope of this review.

Arteriogenesis refers to the formation of collateral vessels from preexisting vessels, triggered by changes in hemodynamic pressure due to distal arterial occlusion [10]. These collaterals can be observed with iodinated contrast beyond the level of arterial occlusion during angiogram procedures. As the arterial lumen narrows due to progressive atherosclerotic stenosis, increased in fluid shear stress remodels the pre-existing artery–arteriolar connections, facilitating blood flow along the path of least resistance [73]. The increase in fluid shear stress enhances the activity of endothelial nitric oxide synthase (eNOS), leading to the release of nitric oxide and promoting the relaxation of smooth muscle cells (SMCs) and vasodilation [74].

Alongside, *VEGF* is released with monocyte chemotactic protein-1 (MCP-1), which facilitates the upregulation of cell adhesion molecules (CAMs) on the endothelial cell surface and recruitment of monocytes, respectively [75]. Monocytes and platelets localize to the CAMs where they secrete various growth factors and cytokines to stimulate endothelial cell proliferation, induce a switch in SMCs from a contractile to a proliferative phenotype, and ultimately promote the proliferation of collateral arterioles [74]. The process concludes with collateral vessel pruning, where many smaller arterioles occlude in favor of fewer, larger arterioles, optimizing flow and distal perfusion [74,76]. However, arteriogenesis often falls short of restoring adequate distal perfusion, as seen in cases of PAD. Collaterals formed through arteriogenesis are frequently observed in patients undergoing surgical intervention with chronic limb-threatening ischemia (CLTI) [77].

While the initiation of arteriogenesis is triggered by increased fluid shear stress, angiogenesis is initiated by tissue ischemia itself [71,78]. Angiogenesis involves the formation of new capillaries in response to ischemia, enhancing the delivery of oxygen and nutrients to the tissue [72,79]. HIF proteins play a crucial role in angiogenesis, as hypoxia stabilizes the alpha subunits, promoting their translocation to the nucleus, heterodimerization with the beta subunit, and DNA binding, followed by the upregulation of numerous potent pro-angiogenic genes [10,80]. Angiogenesis can occur through two mechanisms: sprouting and non-sprouting, or intussusceptive angiogenesis [79,81].

HIF expression can be upregulated in many cell types in the presence of ischemia, including fibroblasts, cardiomyocytes, skeletal muscle cells, immune cells, and solid tumor cells [80,82]. VEGF, the most extensively studied and potent stimulator of angiogenesis in the ischemic microenvironment, is directly upregulated by HIF [10,83]. In sprouting angiogenesis, VEGF binds its receptor VEGFR-2 on endothelial cells which inducing the formation of endothelial tip cells [76,83]. These tip cells guide the growing vessel towards its chemotactic source through their projections, rather than by elongating the blood vessel.

The close interplay between *VEGF* and anti-angiogenic Notch signaling facilitates coordinated formation of the new vessel [77,78]. The tip cell exhibits high *VEGF/VEGFR-2* and high delta like ligand 4 (*Dll4*) expression with low Notch signaling [78]. The increased *Dll4* increases Notch signaling in neighboring endothelial cells, inhibiting their migration. These endothelial cells, with higher Notch signaling and lower *Dll4* expression, form the stalk cells, which exhibit a proliferative phenotype that aids in the elongation of the new vessel [79,80].

HIF signaling in stalk cells maintains a sustained glycolytic metabolism, promoting cellular proliferation in low oxygen tension [84,85]. Additionally, HIF-1α promotes the secretion of matrix metalloproteases (MMPs), urokinase plasminogen activator (uPA), and plasminogen activator inhibitor-1 (PAI-1), which function to degrade the basement membrane and surrounding extracellular matrix (ECM) components, creating space for new blood vessels to form [82]. As the lumen of the new vessel forms through a process known as tubulogenesis, HIF-2α upregulates the expression of *VE-cadherin* to form new endothelial cell junctions, thereby promoting vascular integrity and preventing luminal collapse [80,86].

Additionally, HIF-1α recruits pericytes to surround the endothelial cells, enhancing the vessel’s structural integrity to the vessel and preventing leakage [87]. Furthermore, the delayed onset of HIF-2α compared to HIF-1α elucidates their complementary roles in angiogenesis. HIF-1α rapidly upregulates *VEGF* expression to initiate angiogenesis, while HIF-2α sustains the pro-angiogenic response in chronic hypoxia, promoting vascular remodeling and integrity [72,84].

HIF-1α also plays a key role in recruiting hematopoietic and endothelial progenitor cells (EPCs) from the bone marrow to ischemic tissue sites of by directly upregulating its downstream target, stromal-derived factor 1-alpha (*SDF-1α*) [88,89]. SDF-1α, a cytokine secreted by ischemic tissue cells, enters the peripheral circulation and mobilizes to the bone marrow, where it binds with its receptor, CXCR4, on the cell surface of EPCs [90]. SDF-1α works synergistically with other pro-angiogenic mobilizing factors such as VEGF, hepatocyte growth factor (HGF), and eNOS, to mobilize EPCs from the bone marrow into the peripheral circulation [88,91].

A concentration gradient of SDF-1α is established between the ischemic sites of insult and the peripheral circulation, facilitating the homing of EPCs to ischemic areas [92]. Once there, EPCs proliferate and differentiate into mature endothelial, contributing to the development of new blood vessels [89,90]. Additionally, EPCs secrete various growth factors, including VEGF and SDF-1α, which promote angiogenesis and further recruit EPCs to sites of ischemia [89,91]. Studies have shown that SDF-1 levels rise following ischemic events, and cleavage-resistant gene delivery platforms of SDF-1 show therapeutic potential in rodent models of myocardial infarction [93,94,95].

The HIF proteins upregulate a wide array of known pro-angiogenic genes. A full list can be seen in Table 1.

The HIF proteins, while mostly known to upregulate pro-angiogenic genes, also upregulate a limited repertoire of anti-angiogenic genes. This is thought to be a negative feedback system that interacts with pro-angiogenic genes upregulated by HIFs. A full list can be seen in Table 2.

### HIF-1α vs. HIF-2α

Although HIF-1α and HIF-2α share substantial sequence homology and heterodimerize with HIF-1β to regulate hypoxia-responsive genes, they display distinct temporal kinetics and target gene preferences that define complementary biological functions. HIF-1α acts as an acute responder to hypoxia, rapidly accumulating within minutes of oxygen deprivation and peaking within several hours. It preferentially upregulates glycolytic and angiogenic genes, such as *VEGF*, *GLUT1*, *LDHA*, and *PDK1*, promoting metabolic adaptation to anaerobic conditions and initiating endothelial sprouting [1,103,104]. In contrast, HIF-2α becomes predominant under chronic or sustained hypoxia, showing slower degradation kinetics due to reduced hypoxia-associated factor (HAF)–mediated ubiquitination and maintaining transcriptional activity for 24–48 h [96,97,98]. HIF-2α selectively regulates genes involved in vascular remodeling and oxygen transport, including *VE-cadherin*, *ANGPTL4*, *EPO*, *SOD2*, and *Tie2*, thereby supporting endothelial stability, erythropoiesis, and long-term tissue adaptation [105,106].

The functional distinction between HIF-1α and HIF-2α is reinforced by their different coactivator interactions: HIF-1α primarily associates with CBP/p300, whereas HIF-2α engages PGC-1α and other cofactors that extend transcriptional persistence [107,108]. Together, these differences support a coordinated “handoff” model in which HIF-1α initiates the angiogenic response and metabolic reprogramming, whereas HIF-2α sustains vessel maturation and oxygen homeostasis during prolonged ischemia. Therapeutically, concurrent or sequential activation of both isoforms may produce synergistic benefits: HIF-1α–driven neovascular initiation coupled with HIF-2α–mediated vessel stabilization and erythropoietic support—thereby enhancing the efficacy and durability of angiogenic therapies for ischemic cardiovascular and peripheral arterial diseases.

## 5. HIFs and Ischemic Cardiovascular Disease

Ischemic cardiovascular disease is the leading cause of death in the United States [109]. Over time, the development of atherosclerotic plaque burden leads to arterial stenosis and subsequent occlusion, resulting in downstream tissue ischemia and hypoxia. This condition is marked by reduced blood flow and an inadequate oxygen supply that fails to meet oxygen demand [110]. Atherosclerotic stenosis and occlusion are the pathological basis for many cardiovascular diseases including coronary artery disease (CAD), cerebral ischemia and stroke, mesenteric and renal ischemia, and PAD of the extremities [3,110]. Prolonged and worsening tissue hypoxia from severe atherosclerotic disease ultimately leads to end organ dysfunction, such as ischemic cardiomyopathy in CAD and tissue loss in CLTI, the most severe form of PAD [3]. Additionally, acute plaque rupture and vessel thrombosis in the coronary, peripheral, or cerebral circulation result in acute severe hypoxia and tissue infarction, manifesting as myocardial infarction, acute limb ischemia, and stroke, respectively [111]. Moreover, myocardial conditions such as atrial fibrillation and left ventricular aneurysm, along with atherosclerotic aortic or carotid artery disease, can predispose patients to embolic events, leading to acute tissue ischemia and infarction [109,110]. The abrupt onset of tissue ischemia is often more catastrophic due to the lack of vessel collateralization that can be seen with chronic stenosis and occlusion [3]. This section will focus on the role of HIF-1α in CAD and PAD, followed by therapeutic implications for promoting angiogenesis and vessel collateralization.

As previously discussed, HIF-1α is the major driver of hypoxia-induced angiogenesis and vessel collateralization to ischemic cardiomyocytes due to coronary artery atherosclerosis [3,112]. Many patients with CAD present with vessel collateralization bypassing obstructive plaque, while others lack collaterals. Increased collateralization correlates with reduced infarct size, lower heart failure risk, and decreased mortality [113,114]. In a porcine model of acute myocardial infarction, overexpression of HIF-1α resulted in increased myocardial perfusion post-injury [115]. HIF-1α expression also supports cardioprotection, reduced infarct size, and ischemic preconditioning [116]. In the acute phase of ischemic insult, this HIF-1α–mediated response serves as a protective mechanism to rescue injured tissue and restore perfusion. However, when hypoxic and ischemic insults are prolonged or overwhelming, the compensatory capacity of HIF-1α becomes maladaptive, tipping the balance toward pathological remodeling, chronic inflammation, and disease progression (Figure 2). Hypoxia-inducible factors (HIFs) exhibit a dual role in ischemia–reperfusion injury. While HIF activation promotes angiogenesis and tissue oxygenation, its activation during reperfusion can exacerbate tissue damage by inducing pro-inflammatory signaling pathways and increasing reactive oxygen species (ROS) production [117,118]. These effects contribute to oxidative stress, trigger apoptotic cell death, and amplify inflammatory responses, thereby worsening injury despite HIF’s beneficial functions.

Single nucleotide polymorphisms (SNPs) in the *HIF-1* gene, particularly those resulting in a Pro582Ser substitution, are linked to a reduced collateral formation in coronary artery disease (CAD) and are associated with a clinical presentation of stable exertional angina rather than acute myocardial infarction, suggesting a potential role in earlier disease presentation [119]. In a Mexican population, the SNP rs2057482 is correlated with a decreased risk of developing premature CAD [120]. Although beyond the scope of this review, the same SNP is associated with an increased risk of various cancers and is predictive of clinical outcomes, with reduced binding to microRNA-199a, a negative regulator of HIF-1 levels that binds to the 3′-UTR [121,122] This suggests that elevated HIF-1 protein levels may offer protection against coronary ischemic events but could predispose patients to cancer progression, potentially mediated by microRNA-199a. Indeed, the genetic diversity of *HIF1A* and the varying risks of cancer versus CAD protection are intriguing and warrant further investigation.

Conversely, a recent systemic review and meta-analysis by Chaar and colleagues have found no association between SNPs of HIF-1 and the risk of peripheral artery disease [123]. These risk factors are linked to decreased HIF-1α expression, which reduces VEGF levels and endothelial progenitor cell recruitment. HIF-1α transcriptional activity promotes endothelial cell sprouting, migration, and proliferation under hypoxic conditions [3]. Vascular smooth muscle cells also contribute to vascular integrity during peripheral arterial perfusion [124]. Borton et al. demonstrated that smooth muscle-specific deletion of *ARNT* (HIF-1β) increased vascular permeability and tissue damage in mice after femoral artery ligation, resembling acute limb ischemia [124]. These findings complicate the development of effective HIF-based therapies for PAD.

Acute limb ischemia, often resulting from emboli, differs from chronic PAD but can manifest in PAD patients as acute on chronic limb ischemia. Tuomisto et al. observed elevated expression of HIF-1α, HIF-2α, VEGF, VEGFR-2, and TNF-α in cases of acute on chronic limb ischemia compared to chronic limb ischemia [125]. The heterogeneity among PAD patient populations, including socioeconomic factors, may affect HIF-1α expression and collateralization [126].

## 6. Clinical Limitations of Conventional Revascularization: Rationale for HIF-Based Therapeutic Approaches

The standard treatment for ischemic cardiovascular disease involves restoring arterial perfusion to alleviate hypoxia. In cases of CAD and MI, this is typically accomplished through percutaneous coronary intervention (PCI) using balloons and drug-eluting stents [127]. Some patients with multivessel disease or unfavorable anatomy are better candidates for coronary artery bypass grafting (CABG), which traditionally requires sternotomy and cardiopulmonary bypass, although less invasive options are emerging [128]. Ischemic stroke is treated with tissue plasminogen activator (tPA) or mechanical thrombectomy to restore perfusion [129].

Chronic limb-threatening ischemia (CLTI) is characterized by ischemic pain or tissue loss, most often in the distal lower extremities. Without intervention, these patients face a 22% annual risk of major limb amputation [130]. As with CAD, treatment involves endovascular or surgical revascularization to improve distal blood flow, oxygen delivery, pain relief, wound healing, and limb salvage. However, many patients are not candidates for revascularization due to comorbidities, previous failed interventions, or lack of suitable outflow targets.

Diabetes frequently coexists with CLTI and contributes to both macrovascular and microvascular disease [131]. Occlusions often occur in the tibial and foot arteries, making surgical bypass challenging and less durable due to their distal location. Even when large vessels are successfully treated, microvascular disease in the diabetic foot remains a barrier to healing. Patients who cannot undergo revascularization are deemed to have “no-option” CLTI. In this context, targeting molecular pathways that enhance tissue oxygenation, such as upregulating hypoxia-inducible factor 1-alpha (HIF-1α) to promote angiogenesis, emerges as a promising strategy [132]. By addressing microvascular insufficiency and stimulating neovascular growth, HIF-based therapies offer a novel clinical rationale for improving outcomes in patients who cannot benefit from standard perfusion restoration.

## 7. Prolyl Hydroxylase Domain Inhibition

HIF-1α and HIF-2α are regulated by oxygen-dependent prolyl hydroxylase domain (PHD) enzymes, which target them for degradation. Inhibiting PHD enzymes stabilizes HIF proteins and may promote angiogenesis [133]. Several preclinical studies have shown promise in this approach. In murine hindlimb ischemia models, PHD knockout or knockdown improved perfusion, motor function, and capillary density [134]. Studies using short hairpin RNA (shRNA) targeting PHD2 delivered via a minicircle vector (MC-shPHD2) achieved greater transfection efficiency, higher skeletal muscle HIF-1α levels, and up to 50% blood flow recovery compared to conventional vectors [135,136]. These findings underscore the significance of delivery methods in gene-based therapies.

In myocardial infarction models, *Phd2* knockout led to significantly increased HIF-1α and VEGF levels in peri-infarct tissue, leading to enhanced neovascularization, reduced fibrosis, and improved cardiac function [116,137,138]. Dual knockdown of PHD and FIH further augmented angiogenesis, progenitor cell recruitment, and reduced apoptosis, with upregulation of downstream genes such as *VEGF*, *FGF2*, and *KDR* [139]. Similar cardioprotective effects have been observed in various mouse and human tissue models using pharmacologic or genetic silencing of PHD proteins [140]. However, not all findings have been favorable. In vitro treatment of human endothelial cells with dimethyloxalylglycine (DMOG), a chemical PHD inhibitor, reduced endothelial proliferation, migration, and tube formation, despite increased HIF levels [141]. Pharmacologic PHD inhibitors, such as DMOG, increase HIF levels by preventing HIF-α degradation; however, their effects on endothelial proliferation can be paradoxical. Elevated HIF stabilization may induce the expression of genes that inhibit cell cycle progression or promote differentiation rather than proliferation [142,143,144]. This suggests that the method of HIF stabilization, cell type, and experimental context significantly influence angiogenic responses. Additionally, off-target effects or metabolic changes induced by DMOG could impair endothelial cell growth, despite increased HIF signaling. These dual and sometimes opposing effects may contribute to the limited clinical efficacy of PHD inhibitors in PAD. Further mechanistic studies are needed to delineate these complex pathways and optimize therapeutic strategies, including the exploration of alternative delivery methods and the identification of patient subgroups most likely to benefit.

Clinically, translation of treatments for PAD patients has been limited. A randomized trial using an oral PHD inhibitor GSK1278863 in PAD patients failed to improve walking performance or increase expression HIF-1 target genes [145]. The study faced limitations such as a short treatment duration, oral administration, and lack of angiographic assessment. While oral PHD inhibitors like Roxadustat, Daprodustat, and Vadadustat have been approved to stimulate erythropoiesis in chronic kidney disease, their efficacy in promoting angiogenesis for PAD or CLTI remains unproven. Roxadustat has shown some promise in increasing hemoglobin levels in heart failure patients, which could theoretically increase perfusion in PAD, but direct comparisons have yet to be studied [146]. Additionally, safety concerns, including the risks of thromboembolism and pulmonary hypertension, further complicate their use.

## 8. HIF-1α Gene Overexpression

While inhibiting the inhibitor of HIF-1α is a strategy to promote HIF-1α protein stabilization, inducing HIF-1α overexpression is an alternative to promote neovascularization [132]. Gene therapy for therapeutic angiogenesis employs plasmids or viral vectors to deliver target genes to ischemic tissue, with viral vectors including adenovirus, adeno-associated virus, and retroviruses [146]. Early preclinical studies targeting downstream elements of HIF-1α, such as *VEGF*, *HGF*, and *FGF* showed promise; however, clinical trials involving these growth factors yielded inconsistent results in PAD and CAD [147,148]. A trial in diabetic patients with no-option CLTI using VEGF/HGF bicistronic plasmid therapy reported increased serum VEGF, ABIs, and vessel collateralization, along with improved rest pain [149]. However, the trial was limited by a small cohort. These results suggest that coordinated signaling from multiple factors, as induced by HIF-1α, may be essential for robust angiogenesis.

Xue and colleagues demonstrated in a transgenic diabetes mouse model that cardiomyocyte-specific HIF-1α overexpression increased myocardial capillary density and prevented diabetes-mediated cardiac hypertrophy and glycolytic metabolism remodeling [150]. In a mouse model of myocardial infarction, constitutive expression of HIF-1α attenuated infarct size, increased capillary density, and improved heart function 4 weeks post- myocardial infarction [151]. This supports the rationale for targeting HIF-1α directly rather than its downstream factors. Preclinical studies have reinforced this notion. Intramyocardial injection of HIF-1α/VP16 hybrid increased capillary density and blood flow in rats following LAD occlusion, similar to VEGF treatment [152] Combined HIF-1α and VEGF therapy further increased vessel density but did not reduce infarct size. Remote quadriceps injection of HIF-1α promoted coronary vessel growth, reduced infarct size, and improved ventricular function, suggesting a role in ischemic preconditioning [153]. Sarkar et al. demonstrated in a mouse diabetic model of critical limb ischemia that adenoviral HIF-1α (AdCA5) enhanced arterial remodeling and perfusion, promoting both angiogenesis and arteriogenesis [154]. In diabetic mice, AdCA5 improved perfusion, tissue viability, and motor function and increased circulating angiogenic cells (CACs), which are typically diminished in diabetes [154].

Despite promising preclinical data, clinical trials involving intramuscular HIF-1α gene therapy for PAD have yielded disappointing results. A Phase 1 trial showed safety without tumorigenesis or ocular neovascularization, and some patients experiencing pain resolution and ulcer healing [148]. However, a larger double-blinded, randomized control trial in patients with intermittent claudication showed no improvement in walking time, ABIs, or biomarkers [155]. These results may be attributed to low transfection efficiency. Newer vectors like AAV2 and AAV9 may improve outcomes, although they have not yet been tested in humans. Additionally, inadequate preclinical models and variations in patient pathophysiology further complicate the translation [132].

## 9. Cell-Based HIF-1α Therapies

Stem cell-based therapies hold promise for treating ischemic cardiovascular disease. MSCs, ADSCs, EPCs, and iPSCs can differentiate into various cell types and secrete angiogenic factors [156]. MSCs are particularly appealing due to their ease of harvest and low immunogenicity. Extracellular vesicles (EVs) from stem cells, such as exosomes, deliver pro-angiogenic molecules and influence target cells through paracrine signaling [156]. Stem cells also promote EPC homing via SDF-1α and can differentiate into relevant vascular and cardiac cells [156]. However, the clinical application of unmodified stem cells is limited by poor viability, retention, and homing. Hypoxic/ischemic environments, particularly in diabetics, impair stem cell survival. Strategies to address this include genetic modification, chemical and physical surface modifications, and hydrogel encapsulation. HIF-1α plays a central role in many of these enhancements [132].

Hypoxia preconditioning activates HIF-1α and improves stem cell survival, proliferation, and pro-angiogenic activity [157]. A systematic review of hypoxia-conditioned ADSCs showed consistent upregulation of pro-angiogenic markers and viability [158]. Studies using hypoxia-mimicking agents or reduced oxygen tension confirmed these findings in vitro and in vivo [156]. One murine study showed that hi-MSCs enhanced perfusion, vessel density, and HIF-1α/VEGF expression compared to normoxic MSCs [156]. Direct HIF-1α overexpression in stem cells using plasmids or viral vectors also enhances pro-angiogenic function [159]. CSCs may outperform MSCs in this regard. A study using HIF-1α-transfected CSCs embedded in fibrin gel (HIF-CSC-Gel) improved limb perfusion more than CSCs without the gel [159]. Combined therapy using HIF-1α gene delivery and MSCs in a myocardial infarction model enhanced angiogenesis and cardiac function compared to monotherapies, possibly due to improved MSC engraftment [159]. Future studies should explore combined therapies to optimize outcomes.

Stem cell-derived EVs can also deliver miRNAs like miR-31 and miR-20b to ischemic tissues, promoting angiogenesis and reducing apoptosis in models of myocardial ischemia and reperfusion injury [160]. miR-31 targets FIH, reducing its expression and thereby enhancing HIF-1α activity [161]. Engineering stem cells or EVs with these miRNAs offers another avenue to boost HIF-1α-dependent neovascularization. Interestingly, a recent study shows that overexpressing HIF-1α in MSCs leads to exosomes rich in miRNA-221, capable of shrinking infarct size and cardiac fibrotic scarring in a rat myocardial infarction model [162].

Although HIF-2α’s role in therapeutic angiogenesis has been less explored, it is better understood in cancer, where it aids in vascular remodeling and maintaining integrity during chronic hypoxia [132]. Combining HIF-1α and HIF-2α gene therapies may offer a more durable and functional angiogenic response in atherosclerotic cardiovascular disease [132].

Overall, ischemic cardiovascular diseases, including CAD and PAD, comprise a massive healthcare burden regarding morbidity, mortality, and healthcare economic burden [109]. The mainstay of treatment for both conditions is traditional lifestyle modifications and medical therapy aimed at cardiovascular risk reduction (statins, antiplatelet agents, glycemic control, smoking cessation, etc.) and endovascular or surgical revascularization in cases of AMI and CLTI [127,130]. Surgically or anatomically unfit patients exist in both CAD and PAD, leading to the basis for gene and cell-based therapy to improve neovascularization and improve functional outcomes in cardiac function, maximal walking distance, amputation-free survival, quality of life, morbidity and mortality. Despite the early preclinical promise for both mechanisms of therapy, clinical trials have demonstrated serious limitations of gene- and cell-based therapies, resulting in failed treatment outcomes [148,149]. Hypoxia-inducible factor (HIF) therapies are limited by challenges in drug design, clinical trial execution, safety, and regulatory approval [163]. Additionally, reviews report that HIF inhibitors often lack selectivity and potency, in part because the active sites are elusive and the redundancy between HIF-1α and HIF-2α complicates target validation and drug delivery [164]. A novel drug designed to simultaneously target HIF-1 and HIF-2 could address the redundancy problem that complicates current therapy. Clinical trials have been hampered by poor patient selection, the absence of validated hypoxia biomarkers, and endpoints that fail to capture therapeutic activity in hypoxic environments [163]. Most HIF inhibitors have not progressed beyond early phase trials, resulting in very few approved treatments [165]. Systemic toxicity, off-target effects, and risks such as neoangiogenesis, thrombosis, and cardiovascular events further constrain these approaches. Table 3 summarizes some key human clinical trials evaluating therapeutic strategies for ischemic disease including their delivery challenges and emerging advances.

## 10. Conclusions

HIF-1α is the master regulator of a myriad of pro-angiogenic growth factors and cell survival responses. Consequently, overexpressing HIF-1 is a crucial strategy for enhancing the outcomes of gene and cell-based therapies in treating ischemic cardiovascular diseases. Future gene therapy-based strategies should focus on optimal vector delivery and transfection efficiency, duration of action, and optimal dosing. Hypoxic preconditioning of stem cells has demonstrated improved pro-angiogenic phenotypes across various stem cell types, primarily through the induction of HIF-1α expression. Future clinical studies on cell-based treatment of ischemic cardiovascular disease should incorporate hypoxic preconditioning to optimize pro-angiogenic and cell retention efficacy. Combining gene and cell-based therapies may yield additive effects in promoting stem cell engraftment, neovascularization, and functional outcomes. Additionally, incorporating stem-cell derived exosomes to deliver miRNAs that promote HIF-1α expression is an active area of ongoing preclinical research and could complement the aforementioned therapeutic strategies. The potential of HIF-2α for therapeutic angiogenesis in ischemic cardiovascular disease is not well studied in the current literature, warranting further investigation, either as a standalone therapy or in combination with HIF-1α gene therapy. Novel strategies to increase neovascularization of ischemic tissues which are not necessarily directly related to HIF-1α are under investigation in our research laboratory and involve the membrane-bound adhesion molecule, E-Selectin [162,166,167,168,169,170,171,172,173]. To date, no mechanistic link has been identified between the angiogenic signaling pathways of HIF-1α and membrane-bound E-Selectin. However, the potential for synergism in combination therapies may warrant further study.

## Figures and Tables

**Figure 1 ijms-26-11236-f001:**
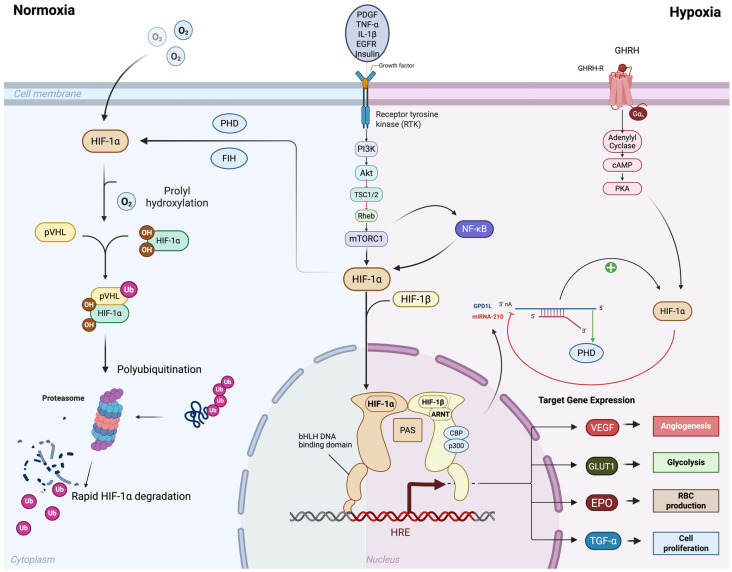
Oxygen-dependent and oxygen-independent regulation of HIF-1α signaling. Under normoxic conditions (**left**), HIF-1α undergoes prolyl hydroxylation by prolyl hydroxylase domain proteins (PHDs) and factor inhibiting HIF (FIH), enabling recognition by von Hippel–Lindau protein (pVHL). This leads to ubiquitination and proteasomal degradation of HIF-1α, preventing transcriptional activity. Under hypoxic conditions (**right**), reduced hydroxylation stabilizes HIF-1α, allowing its accumulation and dimerization with HIF-1β in the nucleus. The HIF-1α/β complex, together with transcriptional co-activators CBP and p300, binds to hypoxia response elements (HREs) to activate transcription of target genes. In addition to this oxygen-dependent regulation, HIF-1α can also be stabilized through oxygen-independent mechanisms, largely mediated by growth factor/receptor signaling pathways. These include PI3K–Akt/mTOR, NF-κB, and EGFR signaling, which enhance HIF-1α synthesis and transcriptional activity even under normoxia. Growth factors such as PDGF, TNF-α, IL-1β, and GHRH further potentiate these effects, amplifying the hypoxic response. Through both oxygen-dependent and oxygen-independent mechanisms, HIF-1α drives the expression of target genes that promote adaptive responses including angiogenesis (*VEGF*), glycolysis (*GLUT1*), erythropoiesis (*EPO*), and cell proliferation (*TGF-α*). Created in BioRender. Reme, A. (2025) https://BioRender.com/wytkie (accessed on 6 October 2025). Abbreviations: HIF, hypoxia inducible factors; PHD, prolyl hydroxylase domain; FIH, Factor Inhibiting HIF; pVHL, von Hippel–Lindau protein; HRE, hypoxia response elements, EGFR, Epidermal Growth Factor Receptor, PDGF, Platelet-Derived Growth Factor; TNF-α, Tumor Necrosis Factor-alpha; IL-1β, Interleukin-1 beta; GHRH, growth hormone-releasing hormone; VEGF, Vascular Endothelial Growth Factor; GLUT1, Glucose Transporter Type 1; EPO, erythropoietin; TGF-α, Transforming Growth Factor Alpha.

**Figure 2 ijms-26-11236-f002:**
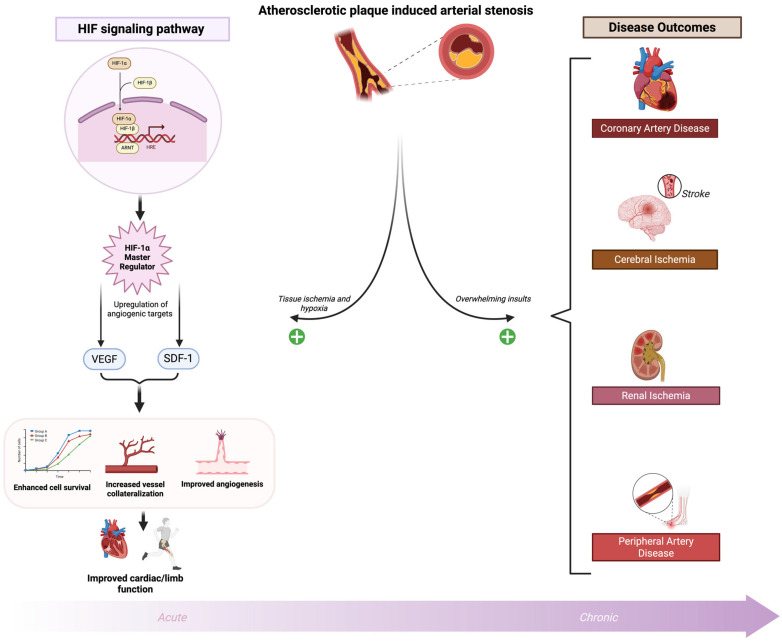
HIF regulation and ischemic cardiovascular diseases. Atherosclerotic plaque–induced arterial stenosis causes tissue hypoxia from cardiovascular and peripheral ischemia. Hypoxia directly upregulates HIF-1α, which activates transcription of angiogenic targets (e.g., VEGF, SDF-1), promoting cell survival, angiogenesis, vessel collateralization, and improved cardiac/limb function. In the acute phase of ischemic insult, this HIF-1α–mediated response serves as a protective mechanism to rescue injured tissue and restore perfusion. However, when hypoxic and ischemic insults are prolonged or overwhelming, the compensatory capacity of HIF-1α becomes maladaptive, tipping the balance toward pathological remodeling, chronic inflammation, and disease progression. Created in BioRender. Reme, A. (2025) https://BioRender.com/wytkieo (accessed on 5 November 2025). Abbreviations: HIF, hypoxia inducible factor; ARNT, aryl hydrocarbon receptor nuclear translocator; VEGF, Vascular Endothelial Growth Factor; SDF-1, Stromal Cell-Derived Factor-1.

**Table 1 ijms-26-11236-t001:** Pro-angiogenic targets of HIF proteins.

Target Gene	Functional Category	Function in Promoting Angiogenesis	Citation
*VEGF* (VEGFA)	Growth factor (cytokine)	Stimulates endothelial cell proliferation, migration, and new blood vessel formation	[9]
*ANGPT1* (Angiopoietin-1)	Growth factor (glycoprotein)	Stabilizes blood vessels and promotes maturation via Tie2 receptor	[93]
*ANGPTL4* (angiopoietin-related protein 4)	Secreted glycoprotein	Regulates vascular permeability and enhances endothelial cell survival	[94]
*PDGFB* (platelet-derived growth factor B)	Growth factor (cytokine)	Recruits pericytes and smooth muscle cells for vessel stabilization	[95]
*FGF2* (Basic fibroblast growth factor)	Growth factor (cytokine)	Promotes proliferation and differentiation of endothelial cells	[96]
*SDF-1* (CXCL12)	CXC- family chemokine	Attracts endothelial progenitor cells to ischemic tissue	[91]
*PIG* (Placental growth factor)	Growth factor (cytokine)	Enhances VEGF-driven angiogenesis and inflammatory cell recruitment	[97]
*EPO* (Erythropoietin)	Glycoprotein hormone	Indirectly promotes angiogenesis by enhancing red blood cell mass and oxygen delivery	[3]
*MMP2*, *MMP9* (Matrix metalloproteinase-2, -9)	Zinc-dependent proteolytic matrix enzyme	Degrades extracellular matrix for endothelial migration and angiogenic sprouting	[98,99]

**Table 2 ijms-26-11236-t002:** Anti-angiogenic targets of HIF proteins.

Target Gene	Functional Category	Function in Inhibiting Angiogenesis	Citation
Regulator of G protein Signaling 5 (*RGS5*)	GTPase-activating protein	Induces endothelial apoptosis and antagonizes VEGF signaling	[100]
Notch1	Transmembrane receptor	Limits endothelial sprouting during angiogenesis by promoting stalk cell morphology	[77,78,79,80]
*DII4* (Delta-like canoical Notch ligand 4)	Transmembrane ligand for Notch1 receptor	Limits endothelial sprouting during angiogenesis by promoting stalk cell morphology	[78]
*TIMP-1*, *TIMP-3* (tissue inhibitor of metalloproteinase -1, -3)	Glycoprotein	Inhibitors and negative regulators of MMP-2 and MMP-9	[101]
*BNIP3* (BCL2/E1B interacting protein 3)	Mitophagy receptor	Promotes mitophagy, and reduces reactive oxygen species production and HIF-1α stabilization	[102]

**Table 3 ijms-26-11236-t003:** Clinical trials of therapeutic strategies in ischemic disease.

Therapy Type	Representative Trial/Model	Delivery Method/Vector	Endpoints Evaluated	Key Outcomes	Delivery Challenges/Limitations	Emerging Advances
PHD Inhibitors	Oral GSK1278863 in PAD patients (Randomized Trial) [145]; Preclinical murine hindlimb and MI models [116,134,135,136,137,138,139,140]	Oral administration; shRNA (MC-shPHD2 minicircle vector)	ABI, perfusion, VEGF expression, cardiac EF	Preclinical: increased perfusion, increased capillary density, decreased cardiac function. Clinical: no ABI or walking improvement.	Oral dosing limits tissue bioavailability; systemic toxicity (thrombosis, pulmonary hypertension); inconsistent HIF stabilization across tissues.	Nanoparticle formulations for local PHD inhibitor delivery; controlled-release systems targeting ischemic zones.
HIF-1α Gene Therapy	AdCA5 adenoviral HIF-1α (CLI mice) [155]; Phase I/II PAD clinical trials [156]	Plasmid DNA, adenoviral vectors, hybrid HIF-1α/VP16, AAV2/9 (preclinical)	ABI, perfusion, limb salvage, EF	Preclinical: increased perfusion, decreased infarct size, increased vessel density. Clinical: safe, modest wound healing, but no improvement in ABI or walking time.	Low transfection efficiency; transient gene expression; immune responses; hypoxic tissue limits vector activity.	Exosome- or nanoparticle-based gene delivery; AAV9 vectors for cardiac tropism; dual HIF-1α + VEGF co-expression.
Cell-Based HIF-1α Therapy	Hypoxia-preconditioned MSCs and ADSCs [158,159,160]; HIF-1α–transfected CSCs in fibrin gel [160]; Exosome/miRNA-based models [161,162,166]	Autologous MSCs, ADSCs, CSCs; genetic modification (HIF-1α plasmid/viral); exosomes enriched with proangiogenic miRNAs	Perfusion, limb salvage, EF, infarct size, vessel density	Increased perfusion, Increased vessel density, decreased infarct size, increased functional recovery; improved MSC survival and engraftment.	Poor cell viability and homing; ischemic microenvironment impairs retention; variability in patient-derived cell quality.	Exosome-based delivery (miR-31, miR-221); hydrogel encapsulation; hypoxia-mimetic preconditioning; combination HIF-1α + HIF-2α approaches.

## Data Availability

No new data were created or analyzed in this study. Data sharing is not applicable to this article.

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
