# Peer review of "Biological Regulation of HIF-1α and Its Role in Therapeutic Angiogenesis for Treatment of Ischemic Cardiovascular Disease"

_ijms, 2025, doi:10.3390/ijms262211236_

Round 1
Reviewer 1 Report
Comments and Suggestions for Authors
The manuscript titled, Biological regulation of HIF-1α and its role in therapeutic angiogenesis for treatment of ischemic cardiovascular disease presents a well-organized, comprehensive review of HIF-1α regulation and its therapeutic potential in ischemic cardiovascular disease (ICD). However, the novel contribution is limited due to the following concerns.
Major Concerns
- The review primarily compiles known mechanisms and pathways, including oxygen-dependent and oxygen-independent processes, as well as gene and cell therapies. However, it lacks a critical synthesis or an identification of gaps in the current field. The content overlaps significantly with several earlier reviews (e.g., Semenza, Cell 2012; Rey & Semenza, Cardiovasc Res 2010; Lee et al., Front Cardiovasc Med 2021). To enhance its impact, the authors could compare the existing limitations in translating HIF-based therapies into clinical practice and suggest innovative mechanistic approaches or delivery methods. The review provides a descriptive overview of the data but frequently falls short on critical analysis. For example, the conflicting effects of pharmacologic PHD inhibitors, such as DMOG decreasing endothelial proliferation, should be explored with a focus on their mechanisms. While clinical trial failures of HIF-1α gene therapy are noted, explanations like immune responses, vector efficiency, or disease heterogeneity are only briefly discussed.
- Although the title highlights HIF-1α, the text occasionally references HIF-2α without providing a thorough comparison. HIF-2α plays distinct roles in chronic hypoxia, such as maintaining endothelial stability and supporting erythropoiesis. Omitting a detailed discussion of its functions reduces the scientific depth. Adding a dedicated subsection (HIF-1α vs. HIF-2α) that compares their gene targets, activation kinetics, and possible synergistic therapeutic benefits would enhance clarity and completeness.
- The therapeutic section (PHD inhibitors, gene therapy, and cell-based therapy) is well-organized but largely narrative. The review could be benefited by summarizing outcomes of key human clinical trials in a comparative table (with endpoints such as ABI, perfusion, limb salvage, or EF improvement), discussing delivery challenges (e.g., vector tropism, dosing, hypoxic microenvironment interference) and incorporating recent advances in nanoparticle-based or exosome-based gene delivery systems.
Minor Concerns
- Overall, the English quality is acceptable, but some parts are verbose and repetitive. For instance, “oxygen-dependent degradation” mechanisms are explained multiple times with similar wording. Sentences like “This result implies direct HIF-1α transcriptional upregulation by NFκB in an oxygen-independent manner” could be made more straightforward for better readability.
- Figure 2 (HIF and disease schematic) lacks clarity in terms of quantification and directionality—arrows could specify positive or negative regulation, and disease outcomes would benefit from clearer labeling. While all abbreviations are spelled out at the end of the manuscript, the figure legends should include the full forms of all abbreviations used in the figures.
- Table 1 presents pro-angiogenic targets but does not include their functional categories, such as growth factors, matrix enzymes, or cytokines. Including these categories would enhance clarity. Furthermore, creating a separate table for HIF's anti-angiogenic targets would provide more valuable insight.
- While thorough, the references predominantly focus on classical studies from the 1990s to 2010s. The review should incorporate recent research (2021–2024), including clinical data on PHD inhibitors such as roxadustat, trials involving HIF-1α–based stem cell therapies, and advancements in exosome-mediated angiogenic treatments.
The English is good, but can be improved.
Reviewer 2 Report
Comments and Suggestions for Authors
This manuscript “ “ Biological regulation of HIF-1alpha and its role in therapeutic angi-2 ogenesis for treatment of ischemic cardiovascular disease” offers a comprehensive and timely review of hypoxia-inducible factors (HIFs), particularly HIF-1α, in ischemic cardiovascular disease. It effectively synthesizes the literature on HIF regulation, angiogenic mechanisms, and emerging therapeutic strategies. The topic is important and the review is well organized, with particular strength in its mechanistic insights. However, several issues should be addressed to improve clarity and consistency.
The title specifies “HIF-1α,” suggesting a narrow both the introduction and later sections indicate that HIF-1α and HIF-2α are discussed in depth. To maintain consistency and accurately convey the scope of the review, the title should be revised to include HIF-2α as well.
Section 6, “HIF-1α Modulation for Therapeutic Angiogenesis,” is somewhat misplaced, as it mainly describes mechanical revascularization methods rather than HIF modulation. This section could instead serve to establish the clinical rationale for HIF-based therapies by outlining the limitations of standard treatments.
Finally, the review emphasizes HIF’s therapeutic potential but overlooks its deleterious role in ischemia–reperfusion injury (inflammatory signaling, reactive oxygen species production). Acknowledging this dual nature—particularly the pro-inflammatory and pro-apoptotic effects of HIF activation during reperfusion—would provide a more balanced and accurate discussion.
Comments on the Quality of English LanguageThe manuscript is generally well-written and scientifically accurate, but readability could be improved. Some sentences are long or complex, and there are minor grammatical issues and awkward phrasings throughout. Careful proofreading to simplify sentence structure and improve flow would enhance clarity and overall presentation.
Round 2
Reviewer 1 Report
Comments and Suggestions for Authors
The authors addressed my concerns with the previous version of the manuscript. I recommend the revised version for publication.